# TRAINING ONE-DIMENSIONAL GRAPH NEURAL NETWORKS IS NP-HARD

**Robert Ganian, Mathis Rocton & Simon Wietheger**
Algorithms and Complexity Group, TU Wien, Vienna, Austria
`rganian@gmail.com, {mrocton,swietheger}@ac.tuwien.ac.at`

## ABSTRACT

We initiate the study of the computational complexity of training graph neural networks (GNNs). We consider the classical node classification setting; there, the intractability of training multidimensonal GNNs immediately follows from known lower bounds for training classical neural networks (and holds even for trivial GNNs). However, one-dimensional GNNs form a crucial case of interest: the computational complexity of training such networks depends on both the graphical structure of the network and the properties of the involved activation and aggregation functions. As our main result, we establish the NP-hardness of training ReLU-activated one-dimensional GNNs via a highly non-trivial reduction. We complement this result with algorithmic upper bounds for the training problem in the ReLU-activated and linearly-activated settings.

## 1 INTRODUCTION

Graph neural networks (GNNs) are, without a doubt, among the best studied and most successful contemporary learning models. Today, GNNs are widely used in a variety of settings, including recommender systems (Wang et al., 2018; Ying et al., 2018), pharmaceutics (Fout et al., 2017; Do et al., 2019; Ghorbani et al., 2022) and fraud detection (Dou et al., 2020). And yet, in spite of significant efforts and interest in the research community, our current understanding of the theoretical foundations of graph neural networks is still in its infancy.

In this article, we concentrate on the crucial task of *training* a GNN, i.e., determining the weights and biases which best match provided training data. We note that the computational complexity of the corresponding NEURAL NETWORK TRAINING problem (NNT) has been extensively studied from the complexity-theoretic point of view: there are numerous recent results proving computational intractability even in highly restrictive settings (Goel et al., 2021; Froese et al., 2022; Froese & Hertrich, 2023; Bertschinger et al., 2023) and exact algorithms targeting the problem (Arora et al., 2018; Boob et al., 2022; Brand et al., 2023). And yet, until now no analogous insights have been available for GNNs. The aim of this article is to change this not only by laying down the foundations for the study of the GNN TRAINING problem, but primarily by presenting a highly non-trivial reduction which establishes the NP-hardness of the problem even for 1-*dimensional* GNNs.

MODEL AND FORMALIZATION. While a highly successful line of empirically-driven research has led to the introduction of hundreds of different variants and adaptations of GNNs to specific settings (Zhou et al., 2020; Kanatsoulis & Ribeiro, 2024; Zhao & Zhang, 2024), for our complexity-theoretic study we will focus on the standard formalization for the (semi-supervised) node-classification model as used, e.g., in the pioneering paper of Kipf & Welling (2017) and the survey of Wu et al. (2021)[1]. We provide an intuitive and high-level description of such GNNs below, while formal definitions are deferred to Section 2.

A graph neural network can be viewed as a graph $\mathcal{G} = (\mathcal{V}, \mathcal{E})$ which performs certain computations over a specified sequence of consecutively processed layers. At each layer $\ell$, each vertex $v \in \mathcal{V}$ computes a vector $H_v^{(\ell)}$ over reals called its *feature vector*; the vector's dimensionality depends on

---

[1]The model we use here is sometimes called Graph Convolutional Neural Networks and is arguably the most prominent GNN model to date.

the layer, and we say that the GNN is $a$-dimensional if $a$ is the maximum feature vector dimensionality over all of the network's layers. While the initial feature vectors $H_v^{(0)}$ are provided, at each other layer $\ell \geq 1$ the network needs to compute $H_v^{(\ell)}$ from the feature vectors at layer $\ell - 1$. In the standard model considered here, this will be done via the application of an *aggregation function* (which aggregates the feature vectors of $v$ and all of its neighbors in layer $\ell - 1$ into a single vector $K_v^\ell$) and an *activation function* (which transforms the aggregated vector $K_v^\ell$ into $H_v^{(\ell)}$). After the final layer is processed, the GNN reads the feature vectors at certain specified vertices; these are typically then passed on for further processing, e.g., to a multi-layer perceptron.

A range of aggregation functions have been considered throughout the literature, with notable examples including SUM, MEAN and SPECTRAL aggregation (Kipf & Welling, 2017). The activation functions used for GNNs are typically the same as those used for neural networks, and in this work we focus primarily on the most widely used ReLU activation function. In line with the literature and previous complexity-theoretic studies on neural network training (Abrahamsen et al., 2021; Brand et al., 2023), we assume that the type of aggregation and activation function is fixed and uniform throughout all layers. However, the parameters of the activation function—specifically, the weight and the bias—will differ from layer to layer, and the task in GNN TRAINING is to compute the weights and biases of each layer which minimize the error on the output feature vectors (i.e., their deviation from the provided training data). While the error may be computed using a variety of error functions, for our main lower-bound result it will be sufficient to use the simple $L_0$ error function, which merely counts the number of mislabeled vertices.

RESULTS AND TECHNICAL CONTRIBUTION. It is rather straightforward to show that a GNN consisting of isolated vertices behaves just like a classical fully-connected neural network with the same activation function, where the number of nodes at each layer is precisely equal to the GNN's dimensionality at that layer (see Proposition 2). Hence, existing lower bounds for NEURAL NETWORK TRAINING on fully-connected networks can be immediately transferred to GNN TRAINING on edgeless graphs; however, such results are only known for networks of high dimensionality (see Section 3) and stem purely from the intractability of training higher-dimensional ReLU functions—without requiring any insights into the structure of the GNN or its aggregation function whatsoever.

On the other side of the spectrum lie 1-dimensional GNNs. For these, one in fact cannot hope to transfer lower bounds from NEURAL NETWORK TRAINING, as the corresponding "purely 1-dimensional" ReLU-activated neural networks can be trained in polynomial time (see Proposition 7 later). Essentially, the 1-dimensional case is the crucial one for ReLU-activated GNN TRAINING (ReLU-GNNT): here both the aggregation and activation functions are easy to deal with in isolation, and the challenge stems from how these interact with each other via the graph structure. As our main result, we prove:

**Theorem 1.** 1-*dimensional* ReLU-GNNT *is* NP-*hard for any* $L_p$ *error function with* $p \in [0, 1)$, *and any of the following aggregation functions:* SUM, MEAN *and* SPECTRAL.

The proof of Theorem 1 relies on an intricate reduction from POSITIVE-1-IN-3-SAT[2], which is first developed for the SUM aggregation function and then adapted to the remaining two. On a high level, the difficulty of the reduction stems from the fact that while the weights and biases have a high degree of freedom at each layer, we need to achieve a correspondence between the training outcome and a solution to our initial SAT instance (where each variable is assigned to be either true or false). We note that it seems difficult—perhaps even impossible—to reuse previously developed gadgets and insights targeting NEURAL NETWORK TRAINING, as there each weight and bias is associated with a single edge of the network while here the weights and biases at a layer are applied globally to *every* vertex in the GNN on that layer.

Essentially, our reduction allows us to associate each variable $x_i$ in the SAT instance with a pair of consecutive layers, and uses special gadgets to ensure that there are precisely two viable choices for weights and biases at these two layers: one corresponding to $x_i$ being true, and the other to it being false. The truth value of $x_i$ is then stored in the form of feature vectors on vertices along dedicated path-like subgraphs. As the feature vectors on these subgraphs are subject to the same weights and biases that are used to encode the truth values of other variables, special care is taken to preserve

---

[2]A variant of 3-SAT where each literal is positive, but each clause requires precisely one true literal to be satisfied; see also the proof of Theorem 1.

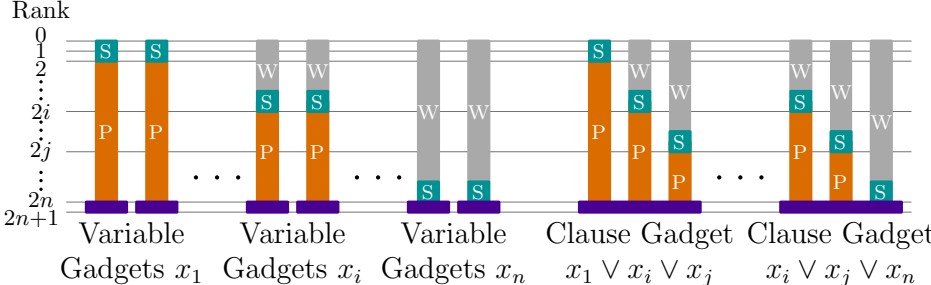

Figure 1: High-level construction for the reduction from POSITIVE-1-IN-3-SAT to ReLU-GNNT. The vertices of the graph are partitioned into $2n + 1$ *ranks* (visualized as horizontal lines). For each variable $x_i$, we construct several identical path-like subgraphs spanning all of the ranks (depicted as multicolored columns): two for the variable itself and an additional copy for each literal of $x_i$ in a clause. The path-like subgraphs for $x_i$ contain a special *Selection gadget* at ranks $2i - 1$ and $2i$ (depicted in blue), where the weights and biases determine whether the feature vector for $x_i$ at all ranks above $2i$ encodes that $x_i$ is set to true, or false. On all other ranks up to $2n$, the path-like subgraphs of $x_i$ consist of specialized *Waiting* and *Preservation gadgets* (illustrated in gray and orange, respectively) which prevent the selection procedure for other variables from interfering with the feature vectors encoding the truth value of $x_i$. On the final rank, we use *Clause* gadgets to ensure that each clause is satisfied by their literals and *Variable* gadgets to ensure that each variable has been correctly set to true or false (depicted in purple).

the truth value of $x_i$ during the whole course of the computation. In particular, our construction partitions the vertices of the instance into *ranks* and ensures that the only "relevant" feature values at layer $\ell$ are precisely the values of vertices belonging to rank $\ell$. In this sense, as the GNN proceeds with the computation, the information can be seen as being "transmitted" along the aforementioned path-like subgraphs.

Once we have forced the GNN to make a choice for each of the variables, we use special gadgets in the final layer to verify that all of the clauses are satisfied by the truth assignment—this is done by ensuring that not satisfying a clause would lead to a specified error threshold being exceeded. A high-level illustration of the reduction is provided in Fig. 1. While the intuition of the reduction is easier to grasp when considering the SUM aggregation function, in our construction we ensure that the obtained graph is uniform—in particular, the degree of each vertex is precisely 6—which allows the proof to work analogously for the other two considered aggregation functions.

Finally, we complement our main theorem with three complexity-theoretic upper bounds.

**Supplementary Result 1.** We obtain the first algorithmic upper bound for ReLU-activated GNN TRAINING. In particular, we show that the training problem on a $\Delta_{\max}$-dimensional GNN with $n$ vertices and $d$ layers can be solved in time at most $L^{\mathcal{O}(1)} 2^{\mathcal{O}(nd\Delta_{\max})} (nd\Delta_{\max})^{\mathcal{O}(d\Delta_{\max}^2)}$, where $L$ is the bit encoding size of the data.

**Supplementary Result 2.** Recall that ReLU-activated GNN TRAINING is computationally intractable even on edgeless graphs (as argued in the beginning of this paragraph), or even for 1-dimensional networks (as shown in Theorem 1). This raises the question: what happens if we combine both of these restrictions? We show that—at least under these severe restrictions and in the exact training setting[3]—GNN TRAINING can be solved in polynomial time.

**Supplementary Result 3.** We establish that exact GNN TRAINING—under the same aggregation and dimensionality conditions as in Theorem 1—becomes polynomial-time solvable when one uses linear activation functions instead of ReLU. While these are far less common than ReLU, we believe this algorithmic result to still be of interest as it follows up on previous complexity-theoretic results for linearly activated neural networks (Panigrahi et al., 2020; Abrahamsen et al., 2021; Brand et al., 2023) and rules out an analogue of Theorem 1 in the exact linearly-activated setting.

---

[3]In exact training, the task is to achieve an exact fit for the training data.

## 2 PRELIMINARIES

Our notation follows prior works on graph neural networks (Kipf & Welling, 2017; Wu et al., 2021) as well as the terminology established for the setting of training neural networks (Froese et al., 2022; Brand et al., 2023). A GNN for semi-supervised node classification with depth $d$ is a tuple $(\mathcal{G}, \Delta, \mathcal{Y})$, where $\mathcal{G} = (\mathcal{V}, \mathcal{E})$ is a graph with adjacency matrix $\boldsymbol{A}$. The vector $\Delta \in \mathbb{N}^{d+1}$ gives the dimensionality of each layer. If $a = \max_{\ell \in \{0, \ldots, d\}} \Delta_\ell$, we call the GNN $a$-*dimensional*. The GNN processes a set of features of every vertex (the *data*), which is given by $X \in \mathbb{R}^{|\mathcal{V}| \times \Delta_0}$. Further, $\mathcal{Y} \subseteq \mathcal{V}$ denotes a set of labeled nodes, and in the training setting we also have a function $\boldsymbol{Y} \colon \mathcal{Y} \to \mathbb{R}^{\Delta_d}$ which determines their target labels.

Given that our work focuses on ReLU- and linearly-activated GNNs, we assume that a trained GNN further comes with weights $W^{(\ell)} \in \mathbb{R}^{\Delta_{\ell-1} \times \Delta_\ell}$ and biases $B^{(\ell)} \in \mathbb{R}^{\Delta_\ell}$ for each layer $\ell \in [d]$, where $[d] = \{1, \ldots, d\}$. The activations (*features*) of the nodes in the $\ell^{\text{th}}$ layer are denoted by $H^{(\ell)} \in \mathbb{R}^{|\mathcal{V}| \times \Delta_\ell}$. In particular, we write $H^{(0)} = X$ for the initial features and $H^{(d)}$ for the final features. Features propagate over the layers. In general, a propagation rule is of the form

$$H^{(\ell)} = f(\boldsymbol{A}, H^{(\ell-1)}, W^{(\ell)}, B^{(\ell)}).$$

*Graph Convolutional Networks* (GCNs), which are the common type of GNNs that are considered in this article, are characterized by a certain form of propagation. There, one first aggregates the features of the node's neighborhood by a given *aggregation function* (such as SUM, MEAN, or MAX) and then applies a given *activation function* (such as ReLU or linear activation) that involves the weights and biases. As common, we will assume a fixed aggregation function and a fixed activation function across all vertices and layers and call the GNN ReLU- or linearly-*activated*, respectively. Formally, the feature of a node $v \in \mathcal{V}$ in layer $\ell \geq 1$ is defined as

$$H_\ell^{(v)} = \text{activation}\left(W^{(\ell)} \cdot \text{aggregation}\left(H_{N[v]}^{(\ell-1)}\right) + B^{(\ell)}\right),$$

where $N[v]$ denotes the closed neighborhood of $v$ in $\mathcal{G}$.

For SUM aggregation and ReLU activation, the $i^{\text{th}}$ dimension in the feature of a node $v \in \mathcal{V}$ in layer $\ell \geq 1$ is computed as

$$(H_v^{(\ell)})_i = \left(\sum_{j=1}^{\Delta_\ell} (W_i^{(\ell)})_j \left(\sum_{u \in N[v]} (H_u^{(\ell-1)})_j\right) + B_i^{(\ell)}\right)^+,$$

where $(x)^+ = \max(0, x)$. The two other types of aggregation functions considered here are MEAN aggregation, where a factor of $\frac{1}{|N[v]|}$ is added before the inner sum, and SPECTRAL aggregation (Kipf & Welling, 2017), which adds a factor of $\frac{1}{\sqrt{|N[v]|}\sqrt{|N[u]|}}$ inside the innermost sum. As for the activation functions, in the final part of our article we will also consider linear activation, where $(\cdot)^+$ is replaced by $(\cdot)$.

The error of a trained GNN is determined by an error function over the predicted and true labels of all labeled data points, where in a real-life network the read-out data points on the final layer would first undergo further processing through a perceptron. For our results, we focus on the special case where the perceptron is trivial (i.e., merely outputs the identity). For $p \in \mathbb{Q}$, the $L_p$ error function considered here assumes we are given predictions $\tilde{y}_1, \ldots, \tilde{y}_k$ and labels $y_1, \ldots, y_k$ and is equal to $\sum_{i \in [k]} ||y_i - \tilde{y}_i||^p$, where $||\cdot||$ denotes the Euclidean norm. Of particular importance will be the case of $p = 0$, which simply counts the number of mislabeled vertices. In particular, we show that the training problem is NP-hard even in this particular setting.

We are now ready to formally state the ReLU-activated GNN training problem considered here.

---

ReLU-GNNT for Aggregation Function $\sigma$ and Error Function $\eta$

**Input:** Graph $\mathcal{G} = (\mathcal{V}, \mathcal{E})$, a $(d+1)$-dimensional vector $\Delta$ of positive integers, data $X$, a set $\mathcal{Y}$ of vertices labeled with $\boldsymbol{Y}$, and an error bound $t$.

**Question:** Are there weights $W^{(1)}, \ldots, W^{(d)}$ and biases $B^{(1)}, \ldots, B^{(d)}$ such that the resulting error on the ReLU-activated GNN using $\sigma$ aggregation is at most $t$?

---

We define the training problem $\text{Lin-GNNT}$ on GNNs with linear activation analogously. Finally, we assume that the bit encoding of the data $X$ is provided on the input and that basic number operations can be carried out in constant time.

## 3 RELATION TO NEURAL NETWORK TRAINING

Before proceeding towards our main result, we first discuss some connections between $\text{ReLU-GNNT}$ ($\text{Lin-GNNT}$) and the corresponding training problems on classical neural networks. For classical neural network training, we follow the standard terminology and definitions used in the literature (Arora et al., 2018; Boob et al., 2022; Froese et al., 2022; Brand et al., 2023) and denote the corresponding training problems as $\text{ReLU-NNT}$ and $\text{Lin-NNT}$, respectively. Below, we formalize an equivalence between training edgeless GNNs and training fully connected neural networks.

**Proposition 2.** *There is a linear-time computable bijection $f$ between (1) the set of $\text{ReLU-GNNT}$ (or $\text{Lin-GNNT}$) instances with $\text{SUM}$, $\text{MEAN}$, or $\text{SPECTRAL}$ aggregation where the graph is $\mathcal{G} = (\mathcal{Y}, \emptyset)$ and (2) the set of $\text{ReLU-NNT}$ (or $\text{Lin-NNT}$) instances where each pair of consecutive layers is fully connected, such that the following holds. For all such GNN-training instances $\mathcal{I} = (\mathcal{G}, \Delta, X, \mathcal{Y}, \boldsymbol{Y}, t)$:*

- *$\mathcal{I}$ is a yes-instance if and only if $f(\mathcal{I})$ is a yes-instance;*
- *$\mathcal{I}$ and $f(\mathcal{I})$ have the same depth, error function, and error bound $t$;*
- *each layer $\ell$ of the classical neural network architecture in $f(\mathcal{I})$ consists of $\Delta_\ell$ nodes.[4]*

As an immediate corollary of the previous proposition, we can transfer known lower bounds where the dimensionality is not bounded by a fixed constant from $\text{ReLU-NNT}$ to $\text{ReLU-GNNT}$. In particular, the following observation builds on the hardness result from Froese et al. (2022).

**Observation 3.** $\text{ReLU-GNNT}$ *with $\text{SUM}$, $\text{MEAN}$, or $\text{SPECTRAL}$ aggregation for any error function $L_p, p \in \mathbb{Q}_{\geq 0}$ is NP-hard, even if the depth is $d = 1$, $\Delta_1 = 1$, and the graph $\mathcal{G} = (\mathcal{V}, \emptyset)$ consists of isolated vertices. Further, the corresponding problems cannot be solved in time $f(\Delta_0) \cdot n^{o(\Delta_0)}$ for any computable function $f$ unless the Exponential Time Hypothesis fails.*

The above reduction works for any fully connected neural network with $\text{ReLU}$-activated nodes, and may hence likely also allow one to transfer other lower bounds from neural network training (Goel et al., 2021; Bertschinger et al., 2023) to the GNNT setting; however, Observation 3 fully suffices for our aim of arguing the intractability of training higher-dimensional graph neural networks.

## 4 TRAINING ONE-DIMENSIONAL GNNS IS NP-HARD

We note that the GNNs obtained in Observation 3 have a dimensionality of 1 in the last layer, but the dimensionality of the first layer is not bounded by any fixed constant. In fact, it is impossible to leverage Proposition 2 to obtain lower bounds for training 1-dimensional GNNs, as the corresponding case in $\text{ReLU}$-activated neural networks boils down to training very simple architectures (see, e.g., Proposition 7 later). Hence, our lower bound for $\text{ReLU-GNNT}$ in the base 1-dimensional setting must exploit both the structure of the network and the properties of the activation function.

For our hardness reduction we require the following observation, which allows us to *shift the absolute values* of weights to the first layer. This lemma is inspired by a similar result on classical neural networks (Brand et al., 2023); curiously, in the classical neural network setting this was used to obtain algorithmic results, while here it facilitates our construction of an NP-hardness reduction. For the remainder of this work, whenever we consider features of dimensionality 1, we simplify the notation by $w^{(\ell)} := (W_1^{(\ell)})_1$ and $b^{(\ell)} := B_1^{(\ell)}$ as well as $H_v^{(\ell)} := (H_v^{(\ell)})_1$.

**Lemma 4.** *Consider any 1-dimensional $\text{ReLU}$-activated GNN with $\text{SUM}$, $\text{MEAN}$, or $\text{SPECTRAL}$ aggregation and depth $d$. For all weights and biases $W, B \in \mathbb{R}^d$, there are weights $\tilde{W}, \tilde{B} \in \mathbb{R}^d$*

---

[4] For convenience, we consider the $0^{\text{th}}$ layer of an classical neural network to describe its input nodes, that is, the input dimensionality of the neural network equals $\Delta_0$.

*such that for all but the first weight in $\tilde{W}$ we have $\tilde{w}^{(2)}, \ldots, \tilde{w}^{(d)} \in \{-1, 1\}$ and using $\tilde{W}$ and $\tilde{B}$ produces the same feature at each vertex after $d$ layers as when using $W$ and $B$.*

With Lemma 4 in hand, we proceed directly to the proof of our main result.

**Theorem 1.** *1-dimensional* ReLU-GNNT *is* NP-*hard for any $L_p$ error function with $p \in [0, 1)$, and any of the following aggregation functions:* SUM*,* MEAN *and* SPECTRAL*.*

*Proof.* We first show the statement for SUM aggregation and $L_0$-error, that is, each mislabeled vertex contributes exactly 1 to the total error. We reduce from the NP-hard POSITIVE 1-IN-3-SAT problem Schaefer (1978): given a set of $n$ variables $x_1, \ldots, x_n$ and $m$ clauses of the form $(x_i \vee x_j \vee x_k)$ (for $1 \le i < j < k \le n$), determine whether there exists an assignment of the variables to {TRUE, FALSE} such that each clause is satisfied by precisely one literal.

We now construct an equivalent 1-dimensional ReLU-GNNT instance of depth $d = 2n + 1$. When describing the construction of the graph $\mathcal{G}$, we will assign each vertex to a "rank"—an integer between 0 and $2n + 1$. One basic building block used in our reduction is a *decision gadget*, which is the subgraph depicted in Fig. 2 that will intuitively be used to *set* a variable $x_i$ to be either TRUE or FALSE.

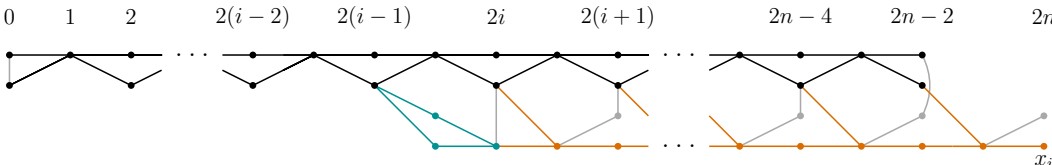

Figure 2: Decision gadget for variable $x_i$. Each column represents a rank. The blue color marks where the *selection* happens and the orange part *preserves* the selected value to the last rank and layer. Gray edges and gray dummy vertices are used to give every vertex except the ones in rank $2n$ (which will be connected to rank $2n + 1$) degree 2 or 4, which will later help us to turn the graph into one where every vertex has the same degree, namely 6, and allow for a more convenient analysis.

We use the above building block in the construction of $\mathcal{G}$ as follows. For each clause $x_i \vee x_j \vee x_k$, we create a copy of each of the three corresponding decision gadgets (i.e., one for $x_i$, one for $x_j$ and one for $x_k$) and connect these together via a *clause gadget* consisting of a *clause check vertex* and a *dummy vertex*, as depicted in Fig. 3(left). Assign the clause check vertex the label 2.

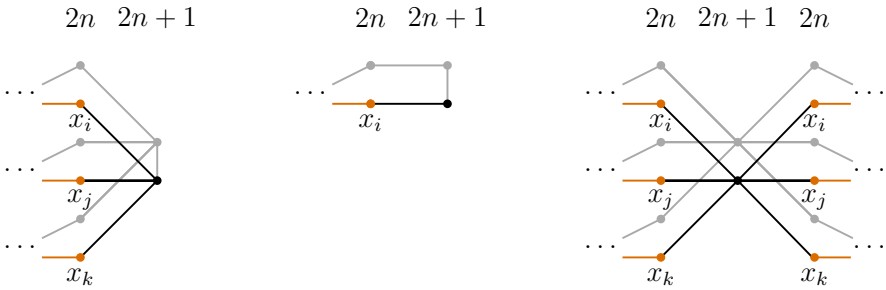

Figure 3: Clause gadget (left), variable gadget (center), and integrity gadget (right). The black vertices are the *clause check*, *variable check*, and *integrity check* vertices, respectively, and are the only labeled vertices in the graph. Gray edges and gray dummy vertices are used to give every vertex degree 2, 4, or 6. Dots indicate that individual copies of the respective decision gadget are included, even if only the ultimate rank is depicted.

We want to ensure that all vertices $x_i$ representing a variable have one of two values in the penultimate layer, which corresponds to the variable being TRUE or FALSE. Thus, for each variable $x_i$, add two *variable gadgets*, each with a *variable check* vertex and a copy of the respective decision gadget, see Fig. 3 (center). Give one of the *variable check* vertices label 1 and the other label 2. Last, pick an arbitrary clause $y_1 \vee y_2 \vee y_3$ and add an *integrity gadget* that connects two decision gadgets for each of its variables (6 in total), see Fig. 3 (right). Give the connecting *integrity check*

vertex label 3. We now define the rank of each vertex. The two left most vertices of each decision gadget are assigned rank 0 and the rank of each other vertex equals its distance to any vertex of rank 0. Note that all labeled vertices (*clause*, *variable*, and *integrity check* vertices) have rank $d = 2n+1$.

While this completes the main part of our construction, we further add a set of *dummy vertices* whose sole purpose is to make the constructed graph 6-regular (a property which will be useful when dealing with the MEAN and SPECTRAL aggregation functions). Note that all vertices have degree 2, 4, or 6. For each vertex $v$ with degree at most 4, add a copy of the clique $K_7$. If $v$ has degree 4, delete an arbitrary edge $e$ from the $K_7$ and add edges from $v$ to the endpoints of $e$. If $v$ has degree 2, delete two disjoint edges from the $K_7$ and add edges from $v$ to the 4 endpoints of these edges. This ensures that $v$ and all added dummy vertices have degree 6.

For the data $X$, let $X_v = 1$ for each vertex $v$ in rank 0 and $X_u = 0$ for every other vertex $u$. Observe that the constructed GNN training problem can be computed in time polynomial in the size of the 1-IN-3-SAT instance. A high-level illustration of the construction for the most important elements is provided in Fig. 1 and the complete graph constructed for an exemplary, simple 1-IN-3-SAT instance is visualized in Fig. 4 in Appendix B. It remains to show that the constructed ReLU-GNNT instance has a solution with error at most $n$ if and only if the 1-IN-3-SAT instance has a solution.

Suppose there is a solution (i.e., a satisfying assignment) to the 1-IN-3-SAT instance and consider the following weights $w^{(\ell)}$ and biases $b^{(\ell)}$, $\ell \in [d]$. Let $w^{(\ell)} = 1$ for all $\ell \in [d]$. For all $i \in [n]$, if $x_i$ is TRUE in the solution, let $b^{(2i-1)} = -1$ and $b^{(2i)} = 0$. Otherwise, let $b^{(2i-1)} = 0$ and $b^{(2i)} = -1$. Let $b^{(2n+1)} = 1$. Crucially, this leads to the following observations: As all biases except for the last one are non-positive and only vertices in rank 0 have a positive feature in $X$, in layer $\ell$ all vertices $v$ with rank $r > \ell$ have $H_v^{(\ell)} = 0$. With this and as the only labeled vertices are in rank $d = 2n + 1$, note that by our construction

- the first time a vertex in rank $r$ may take a positive value is layer $\ell = r$,

- this value is only determined by its neighbors in rank $r-1$ in layer $r-1$ (all other neighbors have feature 0 in layer $r - 1$), and

- whichever value a vertex in rank $r$ takes in layers $\ell > r$ does not matter as this information does not propagate to the labeled vertices in rank $d = 2n + 1$ within the remaining layers.

In particular, this implies that the gray dummy edges and vertices in the gadgets do not impact the final features at the labeled vertices. With these observations, note that the two topmost (black) vertices $v$ in rank $2j, j \in [n]$, in all decision gadgets have $H_v^{(2j)} = 1$, no matter which of the choices are made for the biases $b^{(2i-1)}$ and $b^{(2i)}$ for all $i \leq j$: We prove this by induction over $j$. Clearly, it holds for $j = 0$. Assume for some arbitrary but fixed $j$ we have $H_v^{(2j)} = 1$ for each black vertex $v$ in rank $2j$. If $b^{(2j+1)} = -1$ and $b^{(2j+2)} = 0$, then for the black vertex $v'$ in each gadget in rank $2j + 1$ we have $H_{v'}^{(2j+1)} = 1$ and for each black vertex $v''$ in rank $2(j+1)$ we have $H_{v''}^{(2(j+1))} = 1$. Otherwise, $b^{(2j+1)} = 0$ and $b^{(2j+2)} = -1$, so we have $H_{v'}^{(2j+1)} = 2$ and $H_{v''}^{(2(j+1))} = 1$.

Consider the decision gadgets of any variable $x_i$. Note that in layer $2i-1$ the two bottommost (blue) vertices have exactly one neighbor with feature 1 and all other neighbors have value 0 by the above observations. Thus, if $b^{(2i)} = 0$ and $b^{(2i+1)} = -1$ then for the gadget's bottom-most, blue vertex $v$ in rank $2i + 2$ we have $H_v^{(2i+2)} = 1$ and if $b^{(2i)} = -1$ and $b^{(2i+1)} = 0$ then $H_v^{(2i+2)} = 0$. Next, observe that in the bottom (orange) conservation part, the value is conserved as in every odd layer the feature is increased by 1. This excess is subtracted either in this or the next layer depending on the chosen biases. Thus, each vertex $v$ with even rank $r$ in the bottom path of variable $x_i$ will have $H_v^{(r)} = 1$ if $x_i$ is TRUE and $H_v^{(r)} = 0$ otherwise. This is in particular true for the vertices $x_i$ with rank $2n$. Now consider the last layer $d = 2n + 1$. As we chose a variable assignment satisfying the 1-IN-3-SAT instance, each *clause check* vertex is connected to exactly one vertex $v$ with $H_v^{(d-1)} = 1$; all other neighbors $v'$ have value $H_{v'}^{(d-1)} = 0$. Hence, for each *clause check* vertex $v_c$ we have

$$H_{v_c}^{(d)} = w^{(d)}(1 + 0 \cdot 6) + b^{(d)} = 1 + b^{(d)} = 2,$$

incurring no error. The *integrity check* vertex $v'_c$ does not incur any error as there

$$H^{(d)}_{v'_c} = w^{(d)}(1 \cdot 2 + 0 \cdot 5) + b^{(d)} = 2 + b^{(d)} = 3.$$

Further, exactly half of the *variable check* vertices are mislabeled, giving a total error of $n$ as desired.

For the other direction, suppose there are weights and biases giving a total error of at most $n$. Note that for all $\ell \in [d]$, if two vertices $v, v'$ have isomorphic $\ell$-distance-neighborhoods with respect to the initial features in $X$, then $H^{(\ell)}_v = H^{(\ell)}_{v'}$. In particular, this applies to the final feature of the two *variable check* vertices for each variable. Hence, at most one of these can be labeled correctly. As there are $n$ such pairs of *variable check* vertices, we get that exactly one vertex in each pair is labeled correctly and all *clause check* vertices and the *integrity* vertex are labeled correctly.

Further, observe that every vertex in the graph has the same degree and (except for the ones in rank 0) the same initial value. Hence, for all $r \in [d]$, all vertices in rank $r$ and their adjacent dummy vertices have the same, uniform feature in all layers $\ell < r$ as by construction the non-uniform values from rank 0 only propagate by one rank each layer. For layer $d - 1$, let $a$ be the uniform feature of all labeled vertices in rank $d$ and their adjacent dummy vertices. This further implies that two vertices in rank $r$ with isomorphic $(r-1)$-distance-neighborhood in ranks $r' \leq r$ have the same feature value in layer $r$. In particular, the orange vertex $x_i$ (or $y_i$) in rank $2n$ has the same feature value in layer $2n = d - 1$ across all copies of the decision gadget for variable $x_i$. We hence refer to the feature value of *any* of these vertices in layer $\ell$ by $H^{(d-1)}_{x_i}$. With this, we can narrow down the weight and bias in the ultimate layer. By Lemma 4, we can assume $w^{(d)} \in \{-1, 1\}$. First, consider the case $w^{(d)} = 1$. Both the *integrity check* vertex and the corresponding *clause check* vertex (i.e. the one for the clause over the same variables as used in the integrity gadget) are labeled correctly, so

$$2(H^{(d-1)}_{y_1} + H^{(d-1)}_{y_2} + H^{(d-1)}_{y_3}) + a + b^{(d)} = 3 \quad \text{and}$$
$$H^{(d-1)}_{y_1} + H^{(d-1)}_{y_2} + H^{(d-1)}_{y_3} + 4a + b^{(d)} = 2. \tag{1}$$

Combining the equations gives $H^{(d-1)}_{y_1} + H^{(d-1)}_{y_2} + H^{(d-1)}_{y_3} = 3a + 1$. Applying this back in Eq. (1) yields $b^{(d)} = -7a + 1$. If $w^{(d)} = -1$ we obtain $H^{(d-1)}_{y_1} + H^{(d-1)}_{y_2} + H^{(d-1)}_{y_3} = 3a - 1$ and $b^{(d)} = 7a + 1$. Next, consider any variable $x_i$ and assume $w^{(d)} = 1$. Exactly one of the corresponding *variable check* vertices is labeled correctly, so exactly one of the following two equations is true:

$$H^{(d-1)}_{x_i} + 6a + b^{(d)} = 2, \quad \text{that is,} \quad H^{(d-1)}_{x_i} = a + 1 \quad \text{or}$$
$$H^{(d-1)}_{x_i} + 6a + b^{(d)} = 1, \quad \text{that is,} \quad H^{(d-1)}_{x_i} = a.$$

We let $x_i$ be TRUE in the first case and FALSE otherwise. Similar to Eq. (1), for each clause $x_i \vee x_j \vee x_k$ the corresponding *clause check* vertex enforces

$$H^{(d-1)}_{x_i} + H^{(d-1)}_{x_j} + H^{(d-1)}_{x_k} + 4a + b^{(d)} = 2, \quad \text{that is,} \quad H^{(d-1)}_{x_i} + H^{(d-1)}_{x_j} + H^{(d-1)}_{x_k} = 3a + 1.$$

This holds if and only if exactly one of the variables $x_i, x_j, x_k$ is TRUE giving a yes-instance of 1-IN-3-SAT. The case $w^{(d)} = -1$ works analogously. For all $i \in [n]$ we have that either

$$-H^{(d-1)}_{x_i} - 6a + b^{(d)} = 2, \quad \text{that is,} \quad H^{(d-1)}_{x_i} = a - 1 \quad \text{or}$$
$$-H^{(d-1)}_{x_i} - 6a + b^{(d)} = 1, \quad \text{that is,} \quad H^{(d-1)}_{x_i} = a.$$

We let $x_i$ be TRUE in the first case and FALSE otherwise. For each clause $x_i \vee x_j \vee x_k$ the corresponding *clause check* vertex enforces

$$-H^{(d-1)}_{x_i} - H^{(d-1)}_{x_j} - H^{(d-1)}_{x_k} - 4a + b = 2, \quad \text{that is,} \quad H^{(d-1)}_{x_i} + H^{(d-1)}_{x_j} + H^{(d-1)}_{x_k} = 3a - 1,$$

which holds if and only if exactly one of the variables $x_i, x_j, x_k$ is TRUE.

The only purpose of the $L_0$ error function is to ensure that the error incurred by the *variable check* vertices is minimal if exactly one of the labels is met exactly (and not if the final feature is somewhere in between the labels 1 and 2). Thus, the same construction with an correspondingly updated error bound can be used with any $L_p$ error function for which $p \in [0, 1)$.

Further, on 6-regular graphs, such as the constructed one, the aggregation variants in the statement can be translated to each other by multiplying or dividing all weights by 6. Hence, solving ReLU-GNNT on this instance with SUM aggregation trivially reduces to solving the same instance with MEAN or SPECTRAL aggregation, yielding hardness for these settings as well. □

# 5 ALGORITHMIC UPPER BOUNDS AND TRACTABLE CLASSES

In this section, we obtain three results which can be seen as counterparts to Theorem 1.

SUPPLEMENTARY RESULT 1. We begin by establishing a general algorithmic upper bound for solving ReLU-GNNT.

**Theorem 5.** $\Delta_{\max}$-*dimensional* ReLU-GNNT *with* SUM, MEAN, *or* SPECTRAL *aggregation and any* $L_p$ *error function for non-negative integer* $p$, *depth* $d$ *and a graph of* $n$ *vertices can be solved in time* $L^{\mathcal{O}(1)} 2^{\mathcal{O}(nd\Delta_{\max})} (nd\Delta_{\max})^{\mathcal{O}(d\Delta_{\max}^2)}$, *where* $L$ *is the bit encoding size for the data* $X$.

*Proof.* We begin by exhaustively branching to determine the following information: for each vertex $v$, dimension $i$ and layer $\ell$, whether the ReLU function will flatten the feature of $v$ at dimension $i$ and layer $\ell$, i.e., whether $(H_v^{(\ell)})_i = 0$. This requires an overall branching factor of at most $2^{nd\Delta_{\max}}$, and allows us to concentrate on the case where we assume to have this information available.

For each branch, we construct a set of (in)equalities (i.e., constraints) whose variables represent the set of weights $W$ and biases $B$ in the GNN. In particular, we use a variable for each scalar in every weight and bias, that is, for all $\ell \in [d]$, $i \in [\Delta_\ell]$ and $j \in [\Delta_{\ell-1}]$, we have the variables $(W_i^{(\ell)})_j$ and $B_i^{(\ell)}$. For convenience, we define for each $\ell \in \{0, \dots, d\}$, every vertex $v$, and dimension $i \in [\Delta_\ell]$ a *placeholder* $v_i^{(\ell)}$ which represents a term capturing the value of $(H_v^{(\ell)})_i$; these will be used to state our constraints more concisely. The definition of these placeholders is as follows:

For all $i \in [\Delta_0]$ and each vertex $v$, we define $v_i^{(0)} := (X_v)_i$. We define $v_i^{(\ell)} := 0$ for all places where we branched that $(H_v^{(\ell)})_i = 0$. The remaining placeholders are defined recursively by

$$ v_i^{(\ell)} := B_i^{(\ell)} + \sum_{j=1}^{\Delta_{\ell-1}} \left( W_{i,j}^{(\ell)} \sum_{u \in N[v]} \frac{1}{C} u_j^{(\ell-1)} \right), $$

where we let $C = 1$ for SUM, $C = |N[v]|$ for MEAN, and $C = \sqrt{|N[v]|}\sqrt{|N[u]|}$ for SPECTRAL aggregation. For each $v_i^{(\ell)}$ which our current branch assumes to be flattened, we add a constraint requiring $B_i^{(\ell)} + \sum_{j=1}^{\Delta_{\ell-1}} \left( W_{i,j}^{(\ell)} \sum_{u \in N[v]} \frac{1}{C} u_j^{(\ell-1)} \right) \leq 0$, while for all others we add a constraint requiring $v_i^{(\ell)} \geq 0$.

Let $t$ be the error bound. To complete the construction, we distinguish two subcases. If $p = 0$ (i.e., if the error function merely counts the number of mislabeled vertices), we perform an additional round of branching to determine a set of $|\mathcal{Y}| - t$ vertices which will be labeled correctly, and in each such branch we add a constraint of the form $v_i^{(d)} = Y(v)_i$ for each selected labeled vertex $v$ and dimension $i \in [\Delta_d]$. This yields another branching factor of $\binom{|\mathcal{Y}|}{t} \leq 2^n$. On the other hand, if $p > 0$, we add one constraint $t \geq \sum_{v \in \mathcal{Y}} ||v_i^{(d)} - Y(v)_i||^p$.

Having constructed the set of inequalities described above, we invoke Renegar's Theorem (Renegar, 1992a;b;c) to solve the corresponding instance of the formula in the Existential Theory of the Reals.

Observe that in both cases, weights and biases that label the data as desired yield a solution to the system of (in)equalities and, vice versa, a solution to the system yields such weights and biases. Thus, deciding whether the system has any solution decides the ReLU-GNNT problem. In both cases we have $\mathcal{O}(d\Delta_{\max}^2)$ variables and $\mathcal{O}(nd\Delta_{\max})$ constraints. Note that each constraint has total degree $\mathcal{O}(d)$. Such an instance of the Existential Theory of the Reals can be decided in time in $L^{\mathcal{O}(1)} (nd\Delta_{\max})^{\mathcal{O}(d\Delta_{\max}^2)}$ (Renegar, 1992a;b;c). Combining this factor with the number of branches yields the stated runtime bounds. $\square$

We note that the same approach can be used to solve Lin-GNNT. There, the algorithm skips the branching step and drops all constraints except for those of the form $v_i^{(d)} = Y(v)_i$.

**Corollary 6.** $\Delta_{\max}$-*dimensional* Lin-GNNT *with* SUM, MEAN, *or* SPECTRAL *aggregation, error bound* $t$, *and any* $L_p$ *error function for non-negative integer* $p$, *depth* $d$ *and a graph of* $n$ *vertices*

*can be solved in time $L^{\mathcal{O}(1)}(nd\Delta_{\max})^{\mathcal{O}(d\Delta_{\max}^2)}$ if $p > 0$ and in time $L^{\mathcal{O}(1)}\binom{|\mathcal{Y}|}{t}(nd\Delta_{\max})^{\mathcal{O}(d\Delta_{\max}^2)}$ if $p = 0$, where $L$ is the bit encoding size for the data $X$.*

SUPPLEMENTARY RESULT 2. Next, we proceed with establishing the polynomial-time solvability of exact training for GNNs which are both 1-dimensional and edgeless. Towards this aim, we first tackle the neural network training problem in the corresponding setting.

**Proposition 7.** *For any honest[5] error function, ReLU-NNT with $t = 0$ and precisely one node per layer can be solved in polynomial time.*

Propositions 2 and 7 together yield tractability for 1-dimensional ReLU-GNNT on edgeless graphs.

**Theorem 8.** *1-dimensional ReLU-GNNT with $t = 0$ on edgeless graphs with SUM, MEAN, or SPECTRAL aggregation and any honest error function can be solved in polynomial time.*

SUPPLEMENTARY RESULT 3. Finally, for the simpler linear activation function, we are able to improve the result of Theorem 5 in the setting of exact training and uniform dimensionality to a polynomial time algorithm. First, we provide a *weight shifting* argument that reduces the number of non-trivial weights. For the following, let $\boldsymbol{I}_a$ denote the identity matrix with dimension $a \in \mathbb{N}$.

**Lemma 9.** *Consider any linearly-activated GNN with SUM, MEAN or SPECTRAL aggregation and depth $d$ such that $\Delta_0 = \ldots = \Delta_d$. For all weights and biases $W \in \mathbb{R}^{d \times \Delta_0 \times \Delta_0}, B \in \mathbb{R}^{d \times \Delta_0}$, there are weights $\tilde{W}, \tilde{B}$ such that all for all $\ell \in \{2, \ldots, d\}$ we have $W^{(\ell)} = \boldsymbol{I}_{\Delta_0}$ and using $\tilde{W}$ and $\tilde{B}$ computes the same feature function at each vertex in the final layer as when using $W$ and $B$.*

We are now able to give a polynomial time algorithm for Lin-GNNT instances where all layers have the same dimensionality—including 1-dimensional networks as a special case. Intuitively, once we use Lemma 9 to assume without loss of generality that all but the first weight matrix in the solution are the identity matrix, we can provide a system of linear equations equivalent to our problem and solve it in polynomial time.

**Theorem 10.** *Lin-GNNT with SUM, MEAN or SPECTRAL aggregation, $t = 0$, and depth $d$ such that $\Delta_0 = \ldots = \Delta_d$ with any honest error function can be solved in polynomial time.*

## 6 CONCLUDING REMARKS

Our main NP-hardness result can be interpreted as showing that the intractability of GNN training does not stem solely from the inherent difficulty of multidimensional classical neural networks training, and hence one cannot hope to make progress by designing heuristics targeting solely this aspect. Still, we view it as merely the first—albeit critical—step towards understanding the complexity of training GNNs, one which opens us multiple follow-up questions. For instance, can one identify natural classes of network architectures which allow for efficient (i.e., polynomial-time) training in the 1-dimensional (or, more generally, $d$-dimensional for constant $d$) setting? Our Theorem 1 excludes efficient training for planar networks as the instances arising from the reduction are in fact planar, but leaves open the question of whether one can efficiently train networks which are, e.g., acyclic. Another question that remains open is whether ReLU-GNNT for 1-dimensional networks is also in NP, or lies higher in the polynomial hierarchy; here, one can note that the training problem on ReLU-activated classical neural networks was recently shown to be complete for the complexity class $\exists \mathbb{R}$, which lies between NP and PSPACE (Bertschinger et al., 2023).

Another important branch of follow-up question asks for the complexity of training variants of graph neural networks that differ from the one discussed here. These includes varying the GNN's objective (e.g., graph classification instead of node classification), the propagation paradigm (by going beyond message passing), or the employed activation and/or loss functions (e.g., by considering Leaky ReLU or Sigmoid activation, the $L_p$ loss function for $p \geq 2$, or the cross-entropy loss after some perceptron post-processing). We believe our work acts as an important base case in the exploration of these areas.

---

[5]An error function is *honest* if it returns a value of 0 precisely when the network perfectly fits the data (Bertschinger et al., 2023).

ACKNOWLEDGMENTS

The authors acknowledge support by the FWF Science Fund (FWF projects 10.55776/Y1329 and 10.55776/COE12). Mathis Rocton additionally acknowledges support by the ⬛ European Union's Horizon 2020 research and innovation COFUND programme (LogiCS@TUWien, grant agreement No 101034440).

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

# A  APPENDIX: FULL PROOFS

PROOF OF PROPOSITION 2

*Proof.* For every depth $d$ ReLU-GNNT (or Lin-GNNT) instance $\mathcal{I} = (\mathcal{G}, \Delta, X, \mathcal{Y}, \boldsymbol{Y}, t)$ we construct a ReLU-NNT (or Lin-NNT) instance $f(\mathcal{I})$ as follows. Let $\mathcal{N}$ be a fully connected neural network architecture of depth $d$ (that is, $\mathcal{N}$ yields $d$ consecutive propagation steps). Let each layer $\ell \in \{0, \ldots, d\}$ in $\mathcal{N}$ consist of $\Delta_\ell$ ReLU-activated nodes. Let $D$ contain a data point $(X_v, \boldsymbol{Y}(v))$ for every labeled vertex $v \in \mathcal{Y}$. Keep the same error bound $t$ and error function. Note that this construction describes a bijection between the two sets of instances and $f$ as well as its inversion $f^{-1}$ are computable in linear time.

It remains to argue that $f$ preserves yes- and no-instances, which we prove by showing that each set of weights $W^{(1)}, \ldots, W^{(d)}$ and biases $B^{(1)}, \ldots, B^{(d)}$ incurs the same error on $\mathcal{I}$ as on $f(\mathcal{I})$. Fix any such set of weights and biases. Recalling Section 2, let $(H_v^{(\ell)})_i$ be the value of the $i^{\text{th}}$ dimension in layer $\ell$ at vertex $v$, when processing the GNN in $\mathcal{I}$ with these weights and biases. Analogously, for each data point $(x, y) \in D$, each layer $\ell \in \{0, \ldots, d\}$, and all $i \in [\Delta_\ell]$, let $(\tilde{H}_{(x,y)}^{(\ell)})_i$ denote the value of the $i^{\text{th}}$ node in layer $\ell$ in the neural network $\mathcal{N}$ of $f(\mathcal{I})$ when processing data point $(x, y)$ with the fixed weights and biases. Let $g$ be the bijection that associates each vertex $v$ in $\mathcal{I}$ with its associated data point $g(v)$ in $f(\mathcal{I})$. We prove by induction over the layers that $(H_v^{(\ell)})_i = (\tilde{H}_{g(v)}^{(\ell)})_i$ for all choices of $v, \ell$, and $i$. By construction, we have $(H_v^{(0)})_i = (\tilde{H}_{g(v)}^{(0)})_i$. Further, the propagation rules in classical neural networks and GNNs respectively yield that

$$(\tilde{H}_{g(v)}^{(\ell)})_i = \left( \sum_{j=1}^{\Delta_\ell} (W_i^{(\ell)})_j \cdot (\tilde{H}_{g(v)}^{(\ell-1)})_j + B_i^{(\ell)} \right)^+ \quad \text{and}$$

$$(H_v^{(\ell)})_i = \left( \sum_{j=1}^{\Delta_\ell} (W_i^{(\ell)})_j \cdot (H_v^{(\ell-1)})_j + B_i^{(\ell)} \right)^+,$$

as $\mathcal{G}$ is edgeless and hence $N[v] = \{v\}$. By the induction hypothesis we have $(\tilde{H}_{g(v)}^{(\ell-1)})_j = (H_v^{(\ell-1)})_j$ and thus $(\tilde{H}_{g(v)}^{(\ell)})_i = (H_v^{(\ell)})_i$. As this holds in particular for $\ell = d$ and as $g(v) = (X_v, \boldsymbol{Y}(v))$, the error on both instances is the same. The same reasoning holds for linear activation when all $(\cdot)^+$ are replaced by $(\cdot)$. □

PROOF OF OBSERVATION 3

*Proof.* We reduce from ReLU-NNT with depth 1, $n$ input nodes, and 1 ReLU-activated output node. This problem is NP-hard and cannot be solved in time $f(n) \cdot |D|^{o(\Delta_o)}$ for any computable function $f$ unless the Exponential Time Hypothesis fails (Froese et al., 2022). We note that strictly speaking, Froese et al. (2022) show hardness on a depth 2 neural network where the hidden layer consists of 1 ReLU-activated node and the output layer consists of 1 node that is activated by multiplying the value at the hidden node by a weight $a \in \{-1, 1\}$. Nevertheless, their construction immediately also yields the lower bound for the case where the output layer is omitted and the training occurs directly on the ReLU-activated neuron. The NP-hardness for depth 1 ReLU-GNNT then follows by Proposition 2. □

PROOF OF LEMMA 4

*Proof.* We start with the weights $W$ and biases $B$ and iteratively update them to have weights in $\{-1, 1\}$ while preserving the computed features. Let $\ell$ be the last layer with $w^{(\ell)} \notin \{-1, 1\}$. If $\ell = 1$ there is nothing to prove. Otherwise, update the weights and biases by setting

$$\tilde{w}^{(\ell)} := \text{sgn}(w^{(\ell)}), \quad \tilde{w}^{(\ell-1)} := |w^{(\ell)}| w^{(\ell-1)}, \quad \text{and} \quad \tilde{b}^{(\ell-1)} := |w^{(\ell)}| b^{(\ell-1)},$$

where $\text{sgn}(x) = 1$ if $x \geq 0$ and $\text{sgn}(x) = -1$, otherwise. Then replace the respective weights and biases accordingly. The updated weights and biases produce the same features as before: Consider

any vertex $v$ with closed neighborhood $N[v]$. With the original set of weights and biases we have

$$H_v^{(\ell)} = \left( w^{(\ell)} \sum_{u \in N[v]} \frac{1}{C} \left( w^{(\ell-1)} H_u^{(\ell-1)} + b^{(\ell-1)} \right)^+ + b^{(\ell)} \right)^+,$$

where $C = 1$ for SUM, $C = |N[v]|$ for MEAN, and $C = \sqrt{|N[v]|}\sqrt{|N[u]|}$ for SPECTRAL aggregation. For the updated weights and biases we have

$$\tilde{H}_v^{(\ell)} = \left( \operatorname{sgn}(w^{(\ell)}) \sum_{u \in N[v]} \frac{1}{C} \left( |w^{(\ell)}| w^{(\ell-1)} H_u^{(\ell-1)} + |w^{(\ell)}| b^{(\ell-1)} \right)^+ + b^{(\ell)} \right)^+.$$

If $w^{(\ell)} = 0$ we have $H_v^{(\ell)} = \tilde{H}_v^{(\ell)} = (b^{(\ell)})^+$. Otherwise observe that for all $u \in N[v]$ we have $w^{(\ell-1)} H_u^{(\ell-1)} + b^{(\ell-1)} \leq 0$ if and only if $|w^{(\ell)}| w^{(\ell-1)} H_u^{(\ell-1)} + |w^{(\ell)}| b^{(\ell-1)} \leq 0$. Hence, for all $u \in N[v]$,

$$w^{(\ell)} (w^{(\ell-1)} H_u^{(\ell-1)} + b^{(\ell-1)})^+ = \operatorname{sgn}(w^{(\ell)})(|w^{(\ell)}| w^{(\ell-1)} H_u^{(\ell-1)} + |w^{(\ell)}| b^{(\ell-1)})^+$$

and thus $H_v^{(\ell)} = \tilde{H}_v^{(\ell)}$. This procedure can be repeated until $\ell = 1$. $\qquad\square$

PROOF OF PROPOSITION 7

*Proof.* We prove the statement by showing that every 1-dimensional ReLU-NNT instance of depth $d > 3$ can be reduced to an equivalent 1-dimensional instance of depth at most 3. We start by proving by induction over the layers that the function computed by $\mathcal{N}$ on $D$ in layer $\ell$ can be expressed by

$$f^\ell(x) = \tilde{w} \min(k, \max(j, x)) + \tilde{b}$$

for some $j, k, \tilde{w}, \tilde{b} \in \mathbb{R}$ with $j \leq k$. In the input layer $\ell = 0$, the claim holds by letting $\tilde{w} = 1, \tilde{b} = 0, j = \min_{(x_i, y_i) \in D} x_i, k = \max_{(x_i, y_i) \in D} x_i$. Now assume the claim holds in any fixed layer $\ell$. Let the weight and bias for layer $\ell + 1$ be $w^{(\ell+1)}$ and $b^{(\ell+1)}$. Then, by the inductive hypothesis, the function computed in layer $\ell + 1$ is

$$f^{\ell+1}(x) = \left( w^{(\ell+1)} \tilde{w} \min(k, \max(j, x)) + w^{(\ell+1)} \tilde{b} + b^{(\ell+1)} \right)^+.$$

If $w^{(\ell+1)} = 0$, the claim holds for layer $\ell + 1$ by letting $\tilde{w}' = 0$ and $\tilde{b}' = \max(0, b^{(\ell+1)})$. If $\tilde{w} = 0$, the claim holds with $\tilde{w}' = 0$ and $\tilde{b}' = \max(0, w^{(\ell+1)} \tilde{b} + b^{(\ell+1)})$. Otherwise, let

$$\tilde{w}' = w^{(\ell+1)} \tilde{w}, \quad \tilde{b}' = w^{(\ell+1)} \tilde{b} + b^{(\ell+1)}, \quad \tilde{x}_{\text{cut}} = -(\tilde{b} + b^{(\ell+1)}/w^{(\ell+1)})/\tilde{w}.$$

Consider a function $g(x) = (wx + b)^+$ for any $w, b, x \in \mathbb{R}$ and let $x_{\text{cut}} = -b/w$. Note that we have $g(x) = w \cdot \max(x_{\text{cut}}, x) + b$ if $w > 0$ and $g(x) = w \cdot \min(x_{\text{cut}}, x) + b$ if $w < 0$. Thus, we get for all $x$ in $D$ that

$$f^{\ell+1}(x) = w^{(\ell+1)} \tilde{w} \min(k', \max(j', x)) + w^{(\ell+1)} \tilde{b} + b^{(\ell+1)} = \tilde{w}' \min(k', \max(j', x)) + \tilde{b}'$$

where we let $j' = \max(j, \tilde{x}_{\text{cut}})$ and $k' = \max(k, \tilde{x}_{\text{cut}})$ if $w^{(\ell+1)} \tilde{w} > 0$ and $j' = \min(j, \tilde{x}_{\text{cut}})$ as well as $k' = \min(k, \tilde{x}_{\text{cut}})$, otherwise.

Let $\tilde{w}, \tilde{b}, j$, and $k$ be as for $f^d$ in the last layer of $\mathcal{N}$. We now construct a ReLU-activated 1-dimensional trained neural network $\mathcal{N}'$ with depth $d' \leq 3$ that computes $f^d$ on $D$. If $\tilde{w} = 0$, we let $d' = 1, w^{(1)} = 0$, and $b^{(1)} = \tilde{b}$. Otherwise, we require depth $d = 3$. If $\tilde{w} > 0$, let

$$w^{(1)} = \tilde{w}, \quad b^{(1)} = -\tilde{w}j, \quad w^{(2)} = -1, \quad b^{(2)} = \tilde{w}(k - j), \quad w^{(3)} = -1, \quad b^{(3)} = \tilde{b} + \tilde{w}k.$$

Then, $\mathcal{N}'$ computes the function

$$f('x) = \left( -\left( -(\tilde{w}x - \tilde{w}j)^+ + \tilde{w}(k - j) \right)^+ + \tilde{b} + \tilde{w}k \right)^+$$
$$= \left( -\left( -(\tilde{w}\max(j, x) - \tilde{w}j) + \tilde{w}(k - j) \right)^+ + \tilde{b} + \tilde{w}k \right)^+$$

where we expand the innermost $(\cdot)^+$ using the observation on $g(x)$. We further get

$$f'(x) = \left(-\left(-\tilde{w}\max(j,x) + \tilde{w}k\right)^+ + \tilde{b} + \tilde{w}k\right)^+$$
$$= \left(-\left(-\tilde{w}\min(k,\max(j,x)) + \tilde{w}k\right) + \tilde{b} + \tilde{w}k\right)^+$$
$$= \left(\tilde{w}\min(k,\max(j,x)) + \tilde{b}\right)^+.$$

If $\tilde{w} < 0$, let

$$w^{(1)} = |\tilde{w}|, \quad b^{(1)} = -|\tilde{w}|j, \quad w^{(2)} = -1, \quad b^{(2)} = |\tilde{w}|(k-j), \quad w^{(3)} = 1, \quad b^{(3)} = \tilde{b} + \tilde{w}k.$$

We can reuse the above steps except the last one to get that here the computed function is the same:

$$f'(x) = \left((-|\tilde{w}|\min(k,\max(j,x)) + |\tilde{w}|k) + \tilde{b} + \tilde{w}k\right)^+ = \left(\tilde{w}\min(k,\max(j,x)) + \tilde{b}\right)^+.$$

As $\mathcal{N}$ is ReLU-activated and is described by $f^{d+1}(x) = \tilde{w}\min(k,\max(j,x)) + \tilde{b}$ on $D$, we know that $f^{d+1}(x_i) \geq 0$ for all $x_i$ in $D$. Hence, for all $x_i$ in $D$, we have $f'(x) = f^{d+1}(x)$.

As $\mathcal{N}'$ can express the same set of functions on $D$ as $\mathcal{N}$, solving the training problem for $\mathcal{N}$ reduces to solving the training problem on $\mathcal{N}'$ when using the same data set $D$, error function, and error bound. The latter, with depth at most 3 and just one node per layer, can be decided in polynomial time, for example by the algorithm of Brand et al. (2023). We note that while their result is stated for the $L_2$ error function, it also holds for every other honest error function as one honest error function yields error 0 if and only if all other honest error functions do the same. $\qquad\square$

PROOF OF LEMMA 9

*Proof.* We start with the weights $W$ and biases $B$ and iteratively update them to obtain identity matrices for the weights while preserving the computed features. Let $\ell$ be the last layer with $W^{(\ell)} \neq I_{\Delta_0}$. If $\ell = 1$ there is nothing to prove. Otherwise, update the weights and biases by setting

$$\tilde{W}^{(\ell)} := I_{\Delta_0}, \quad \tilde{W}^{(\ell-1)} := W^{(\ell)}W^{(\ell-1)}, \quad \text{and} \quad \tilde{B}^{(\ell-1)} := W^{(\ell)}B^{(\ell-1)}.$$

Then replace the respective weights and biases accordingly. The updated weights and biases produce the same features as before: Let $n$ denote the number of vertices in the graph $\mathcal{G}$, let $\tilde{A} = A + I_n$ be its adjacency matrix with forced self-loops, and let $\tilde{D}$ be its degree matrix that has the vertex degrees along the diagonal and 0s everywhere else. The forced self-loops increase the degree of every vertex by 1. Let $C = C' = I_n$ for SUM, $C = \tilde{D}^{-1}$ and $C = I_n$ for MEAN, and $C = C' = \tilde{D}^{-1/2}$ for SPECTRAL aggregation. With the original set of weights and biases we have

$$H_v^{(\ell)} = W^{(\ell)}(W^{(\ell-1)}C\tilde{A}C'H^{(\ell-1)} + B^{(\ell-1)}) + B^{(\ell)}$$

while for the updated weights and biases we have

$$\tilde{H}_v^{(\ell)} = I_{\Delta_0}((W^{(\ell)}W^{(\ell-1)})C\tilde{A}C'H^{(\ell-1)} + W^{(\ell)}B^{(\ell-1)}) + B^{(\ell)} = H_v^{(\ell)}.$$

This procedure can be repeated until $\ell = 1$. $\qquad\square$

PROOF OF THEOREM 10

*Proof.* For every such Lin-GNNT instance, if there is a solution at all, then there are weights and biases such that all but the first weight matrix are the identity matrix (Lemma 9). Finding these weights and biases corresponds to finding a solution for the following set of linear equations. Let there be a variable $(W_i^{(1)})_j$ for all $i, j \in [\Delta_0]$ and a variable $B_i^{(\ell)}$ for all $\ell \in [d]$ and $i \in [\Delta_0]$. For convenience, we state the equations using a placeholder $v_i^{(\ell)}$ for every $\ell \in \{0, \ldots, d\}$, every vertex $v$, and dimension $i \in [\Delta_0]$. For all $i \in [\Delta_0]$ and each vertex $v$, we define $v_i^{(0)} := (X_v)_i$ and

$$v_i^{(1)} := B_i^{(1)} + \sum_{j=1}^{\Delta_0}\left(W_{i,j}^{(1)}\sum_{u \in N[v]}\frac{1}{C}u_j^{(0)}\right),$$

where $C = 1$ for SUM, $C = |N[v]|$ for MEAN, and $C = \sqrt{|N[v]|}\sqrt{|N[u]|}$ for SPECTRAL aggregation. The remaining placeholders are defined recursively by

$$v_i^{(\ell)} := B_i^{(\ell)} + \sum_{j=1}^{\Delta_{\ell-1}} \Big( \sum_{u \in N[v]} \frac{1}{C} u_j^{(\ell-1)} \Big).$$

For every labeled vertex $v \in \mathcal{Y}$ and $i \in [\Delta_0]$, add the equation $\mathbf{Y}(v) = v_i^{(d)}$. This is a linear equation as, even though the $v_i^{(d)}$ are defined recursively, in none of the expressions two variables are multiplied with each other. As there are $\Delta_0(\Delta_0 + d)$ variables, the system of linear equations can be solved in polynomial time (e.g., by Gaussian elimination). □

# B APPENDIX: ADDITIONAL FIGURE

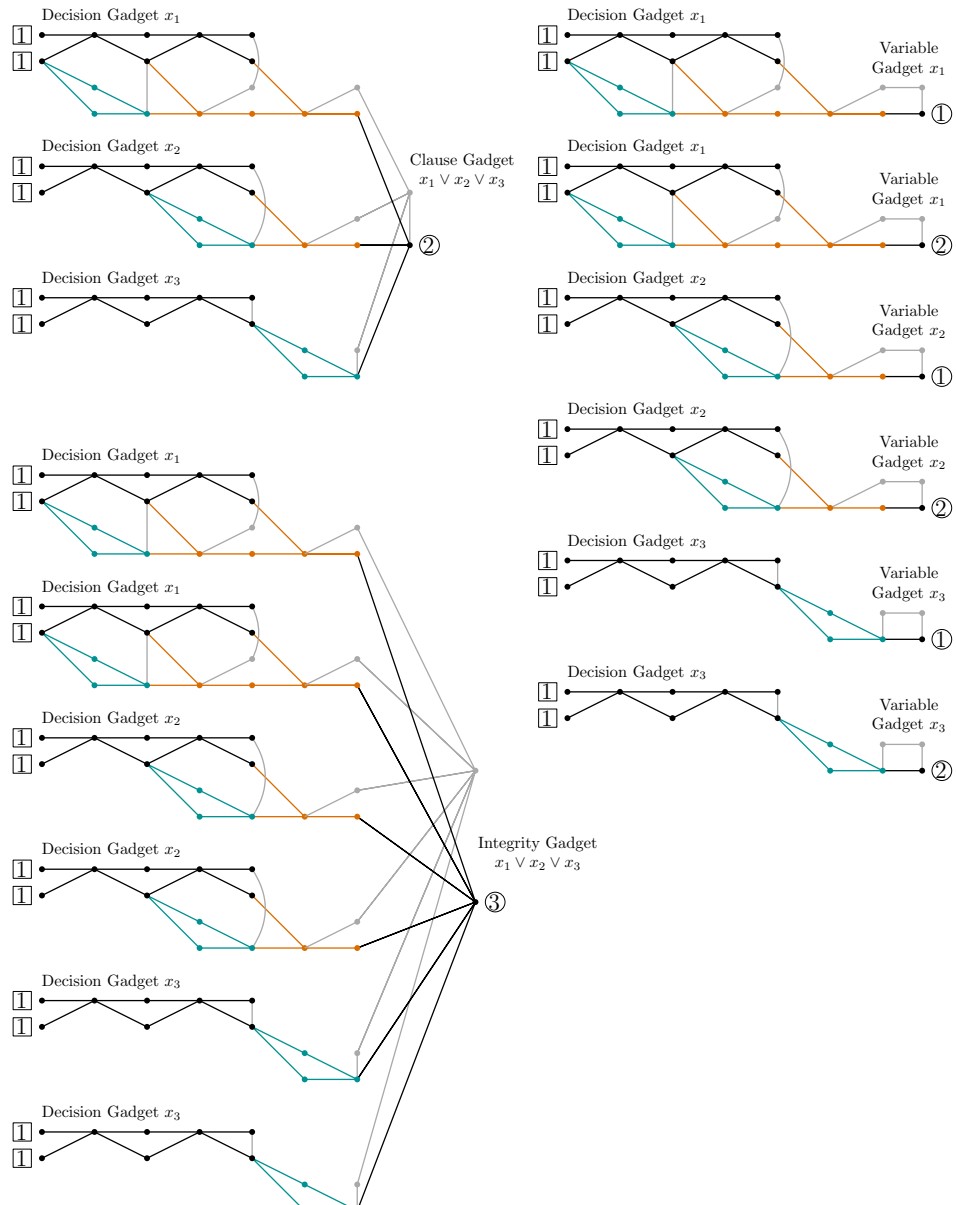

Figure 4: The graph constructed in the reduction for Theorem 1 for a 1-IN-3-SAT instance with variables $x_1, x_2, x_3$ and a single clause $x_1 \vee x_2 \vee x_3$. The initial label (data) for each vertex is 0, except for the leftmost vertices, where it is 1 (marked with a 1 in a square). Target labels are marked with a circle. The error bound is 3, i.e., the number of variables in the 1-IN-3-SAT instance. An exemplary solution that corresponds to setting only $x_1$ to be TRUE is setting $w_i = 1$ for all $i \in \{1, \ldots, 7\}$ and $b_1 = b_4 = b_6 = 0$ and $b_2 = b_3 = b_5 = -1$ and $b_7 = 1$. This way, the rightmost orange vertex of each decision gadget for $x_1$ has value 1 in the penultimate layer. The rightmost orange vertex in decision gadget for $x_2$ and the rightmost blue vertex in the decision gadget for $x_3$ as well as all labeled vertices and their dummy-neighbors have value 0 in that layer. Hence, in the ultimate layer, with the added bias of 1, the labeled clause gadget vertex has value 2 and the labeled integrity gadget vertex has label 3. The solution mislabels only the labeled vertices in the first variable gadget for $x_1$ and the second variable gadgets for $x_2$ and $x_3$, respectively.

