# OpenReview forum: "Training One-Dimensional Graph Neural Networks is NP-Hard"
_ICLR.cc/2025/Conference — ICLR 2025 Poster_

### Official Review · Reviewer_U9xM · 2024-10-28

**Soundness:** 3
**Presentation:** 2
**Contribution:** 3
**Rating:** 6
**Confidence:** 4

**Summary:**

The paper investigate the computational complexity of training GNNs.  While it is straightfoward that training a high-dimensional GNN with a single node is NP-hard, the paper focuses on the less explored case of training multi-node, one-dimensional GNNs. The authors prove that training 1-dimensional GNNs is NP-hard when (i) the loss function is $L_p$ for some $p\in[0,1)$; (ii) the aggregation functions is SUM, MEAN, or SPECTRAL; and (iii) the activation function is ReLU. The proof is an reduction from the NP-hard POSITIVE-1-IN-3-SAT problems.

**Strengths:**

The results are solid and contribute to our understanding of the computational hardness of training GNNs.

**Weaknesses:**

1. The paper should provide a definition of the POSITIVE-1-IN-3-SAT problem when introducing the proof idea in the Introduction.
2. It would be helpful to expand the discussion on the assumptions of the main theorem. For instance, is the problem still NP-hard for more commonly used loss functions, such as cross-entropy or $L_2$?
3. The paper could benefit from a more explicit discussion on how these theoretical results might influence the design of GNN architectures or training algorithms in practical applications.

**Questions:**

Refer to "Weaknesses"

---

> ### Author Response · Authors · 2024-11-19
> **Response to Reviewer U9xM**
>
> **Response to W1**: We have added a footnote explaining the problem in the updated manuscript.
>
> **Response to W2**:  This is the first complexity-theoretic study of training GNNs, and whether Theorem 1 carries over to all loss functions, and in particular L_2, remains an interesting open question. Still, we believe that Theorem 1 settles a crucial first step without which it is unlikely to embark on deeper investigations into the problem’s complexity. We updated the Concluding Remarks to more explicitly address this field of future work.
>
> **Response to W3**: As pointed out in our response to Reviewer tj5Z, one interpretation of our hardness result is that the intractability of training GNNs is already due to the difficulty of handling the communication between nodes; hardness does not stem solely from the inherent difficulty of multidimensional classical neural networks training. Hence, one cannot hope to make progress by designing heuristics targeting solely the latter aspect. Put more simply: our result can be interpreted as an indication that heuristics for training neural networks cannot be “directly” applied to also work for GNNs. The updated version of the manuscript now also addresses this point in the Concluding Remarks.

---

> > ### Comment · Reviewer_U9xM · 2024-11-22
> >
> > Thanks for the response. I will maintain my score.

---

### Official Review · Reviewer_bLnv · 2024-10-30

**Soundness:** 4
**Presentation:** 2
**Contribution:** 3
**Rating:** 6
**Confidence:** 3

**Summary:**

The paper shows that 1-dimensional GNN training is hard. In precise wording, for a $L_p$ norm error, finding the optimal weights and biases that decreases the error between the predicted label and the ground truth label lower than a threshold (decision problem) reduces to 1-in-3 SAT problem (same as SAT but each clause has to be satisfied with only one variable).

The paper is novel in sense that other hardness results for neural networks can be used for applied to multi-dimensional GNNs, but here the authors show the same for 1-dimensional which in NNs there is a polynomial time search for it.

The authors follow by introducing algorithmic upperbounds for special cases of NN training and GNN training.

**Strengths:**

The approach to the problem was really non-trivial. The target reduction problem was unexpected to me and like any other algorithmic lower bound, the reduction requires novel approaches in problem design.

**Weaknesses:**

I assume Observation 3. is just a rephrasing of Theorem 1. If so, why it is repeated?

Although the problem is really interesting, and the work has a lot of novel ideas for tackling the problem, still I am not sure (1) if the paper is presented in a correct subject (e.g. if there is any theoretical study of NNs or some similar categories it belongs to that category where reviewers have better background on complexity study) and (2) I am still unclear about the impact of the work in application. Surely it is a theoretical study and we should not expect direct real-life applications but the study is in a corner case (1-d GNNs) where for more dimensions the results are already showing NP-hardness. It would be better if the authors could discuss the potential implications of this work for GNN training in general.

**Theorem 1.** The reduction is nice, however the intuition behind this reduction is still unclear. Although polynomial reductions are essentially not easily understandable, and are mostly innovative, there are intuitive descriptions that can make the reduction understandable. I still am not sure if the proof is completely correct since most of the parts in the proof are still not understood by me..

A list of questions are provided in "questions" section. I recommend writing the proof (or sections before it) again considering the questions, so an abstract image of this reduction becomes clear.

However the gadgets and a high-level shape of the problem is illustrated, still the designed graph is not clear. It would elevate the paper's quality if the reader could imagine a sample graph for even a very small 1-in-3 SAT problem. The number of layers is also not determined. In case (from lemma 4) the 2-layer GNN is representative of any other GNN, the authors could mention that and for a sample SAT instance, show the weights of the optimal network, and the graph.

**Questions:**

I believe the core part of the paper is Theorem 1, and with the theoretical nature of the work the qualification of the paper mostly boils down to checking the proofs. However the proof, and the theorem statements are unclear. I think answering following questions can help me to understand it more, and also placing these answers in the paper enhances its readability:

1. What does the ranks encode. Why there are 2 ranks per node. How a node is assigned to a rank? The authors just introduced the terminology while leaving out the intuition behind using it. Specifically what does this sentence mean?
> In particular, our construction partitions the vertices of the instance into ranks and ensures that the only “relevant” feature values at layer l are precisely the values of vertices belonging to rank l.

2. In theorem 1, and in general please specify what is the range of the labels. It can be that the labels assumed in theorem 1, range differently compared to the more general setup.

3. Please specify the correct labels of the graph in general. Which nodes should be labeled? which nodes are left without labels?

4. Please specify a simple case of 1-in-3 SAT (e.g. with one or two clauses) alongside the corresponding instance of GNN training so it becomes clear what each gadget is introducing to the problem.

5. Number of the layers are not fixed at the beginning of the problem. From lemma 4 it is understood that a 1, or 2 layer GNN is representative of the overall problem. Is that so? If yes please specify that without loss of generality we can assume 2, layers.

6. (Similar to 5) Why there is no relation between the number of layers and any other element in the corresponding graph. The graph seems to be only a function of the 3-in-1 SAT problem. In that case what is the difference between 2 and any n-layer GNN?

7. Why the rank in integrity gadget is 2n, 2n+1 and then again 2n? How many times a clause is repeated as gadgets? Why in the integrity gadget the grey center node is not connected to the black one?

8. What is the configuration of labels w.r.t. clauses in the SAT problem? Please provide an example SAT and show the graph and model labels alongside the prediction of the model.

9. What is the loss in the decision problem corresponding to a given SAT instance?

---

> ### Author Response · Authors · 2024-11-19
> **Response to Reviewer bLnv**
>
> **Response regarding Observation 3 vs. Theorem 1**: Observation 3 and Theorem 1 apply to different settings. Observation 3 establishes the intractability of training GNNs (even edgeless GNNs) of higher dimensionality and follows almost directly from previous work. Theorem 1 establishes the intractability of training 1-dimensional GNNs, and requires entirely new techniques and insights.
>
> **Response regarding the practical impact of the work**: We kindly refer to the first part of our response to Reviewer tj5Z, where we elaborate on this topic in detail.
>
> **Answer to Q1**: There is one unique rank per node, which is assigned by the distance to a node of rank 0, i.e. the minimum number of edges on a path to such a node (see, e.g., paragraph 2 of the proof of Theorem 1). Nodes of rank 0 are exactly the two left-most (according to Figure 2) nodes of each decision gadget, and these nodes are the only ones which are initialized with value 1 (all the nodes of non-zero rank being initialized with value 0).
>
> The quoted sentence means that (1) Since each vertex has a rank, the vertex set is naturally partitioned into classes of vertices with the same rank. Moreover, (2) in each layer L the features of vertices in rank L are essentially only determined by the weight, bias, and the features of vertices in rank L - 1. This allows for an iterative analysis of how the values propagate from rank 0 to rank 2n+1 over the layers 0 to 2n+1.
>
> **Answer to Q2 and Q3**: The statement of Theorem 1 is a general NP-hardness result for the problem as defined on page 4, where no restrictions on the labels are present. That being said, the proof constructs GNNs with input labels from {0,1} and output labels from {1, 2, 3}; hence, it also establishes NP-hardness for the case where labels only come from these special sets. Regarding which nodes receive which labels, beyond the explanation provided in the proof of Theorem 1 one can now refer to the new Figure 4 for an easier overview.
>
> In all our algorithmic results, we do not assume any kind of restriction on the labels.
>
> **Answer to Q4**: Thank you for the nice suggestion. We have updated the manuscript with an illustrative figure that exemplifies the construction for a simple 1-in-3 SAT instance (Figure 4).
>
> **Answer to Q5 and Q6**: We do not fix the number of layers as part of the problem because we consider the most general version of the problem, where the number of layers is part of the problem input. The number of layers of the GNN in Lemma 4 is not 2, but d. We make no assumption about the number of layers of the GNNs in Lemma 4, nor in the rest of the paper. Theorem 1 produces a GNN training instance with depth 2n+1.
>
> If there is any particular point of confusion regarding the above, please let us know; we will be happy to elaborate further.
>
> **Answer to Q7**: The integrity gadget is depicted this way only because of readability considerations: the elements it connects together are all selection gadgets, connected in the part where they have rank 2n: it could be drawn with all 6 variables on the left. There is exactly one clause gadget per clause in the formula. The integrity gadget (which is unique) mimics one of the clause gadgets (it does not matter which one), but with two copies of each variable instead. The gray center vertex in the integrity gadget is, as all gray vertices, a dummy vertex, whose only purpose is to ensure that the vertices in the gadgets have degree 2, 4 or 6 (and this in order to have a 6-regular graph in the end). Hence, since both center vertices (black and gray) already have degree 6, an edge between them is not needed.
>
> **Answer to Q8**: While this is not easy to describe without repeating the construction presented in the proof of Theorem 1, we have added an example SAT instance and solution features (a "prediction") into Figure 4 and its caption.
>
> **Answer to Q9**:  The loss function of the training problem we consider in Theorem 1 is the L_p norm, for any p\in [0,1[. There is no loss function in the SAT problem, all of its clauses have to be exactly satisfied for the instance to be satisfiable.
>
> In the constructed decision problem with L_0 loss function, there is a solution with loss at most n (the number of variables) if and only if the corresponding SAT instance is a yes-instance. This corresponds to labeling precisely one of the two variable gadget vertices for each variable correctly. For L_p error with p\in ]0,1[, this bound needs to be adjusted accordingly. We are happy to elaborate on our answer upon further request.

---

> > ### Comment · Reviewer_bLnv · 2024-11-21
> >
> > Thank you for addressing my concerns and answering my questions. Here I mention some follow-ups only to understand the paper more. However to this point I am convinced that the paper has nice (theoretical) contribution for which I increase my score to suggest acceptance.
> >
> > Specifically, I appreciate adding Figure 4. The paper is now much more understandable.
> >
> > **Follow-up to Q2, and Q3.** Is the problem with output labels in {1, 2} still NP-hard? I mean is solving the binary graph node classification of the same hardness?
> >
> > **Follow-up to Q5, and Q6.** Does this mean that a single-layer GNN which is AXW + b = y also NP-hard? However, I do not see any flaw in the descriptions still it is slightly far from my intuition. This is since A is fixed and ultimately the problem is a 1-layer NN.

---

> > > ### Author Response · Authors · 2024-11-21
> > >
> > > We are happy that our answers helped provide more clarity and that we could improve our write-up building on your feedback. Thank you very much for reflecting these improvements in your assessment! Regarding the two follow-up questions:
> > >
> > > **Response to the Follow-up to Q2+3**: Theorem 1 only establishes hardness for at least 3 labels, and we did not aim to optimize for having the smallest possible set of labels. That being said, we are very confident that our hardness construction (in Theorem 1) could be adapted to also hold for just 2 labels - specifically, even if the labels are in {1,2}. In particular, this can be done by replacing the current integrity gadget with a different one.
> > >
> > > **Response to Follow-up to Q5+6**: The answer to this question depends on the considered dimensionality.
> > >
> > > For 1-dimensional GNN training, the hardness result of Theorem 1 requires that the number of layers is part of the input and not fixed to any constant. So if the input consists of a 1-dimensional GNN architecture and an integer depth d (i.e., d is the number of layers), the training problem is NP-hard. If, on the other hand, we would have a fixed constant d specifying the number of layers and the input consists merely of a 1-dimensional GNN architecture, the question of whether we can train the architecture in polynomial time is open. For the special case where d=1 (i.e., if we fix the depth to 1), the 1-dimensional GNN training problem can be seen to be in P via a direct LP formulation.
> > >
> > > However, if the dimensionality is not bounded by a constant, Observation 3 establishes hardness even for depth 1. This is due to the hardness of training a single ReLU activated neuron (with high input dimensionality) in a classical neural network as established by [Froese et al. 2022].

---

### Official Review · Reviewer_onPv · 2024-11-01

**Soundness:** 3
**Presentation:** 2
**Contribution:** 2
**Rating:** 6
**Confidence:** 3

**Summary:**

The main result is showing NP-hardness for training a 1-dimensional GNN (i.e. the input dimension and width are 1). The proof uses a reduction from the positive-1-in-3-SAT problem. Several other results are given such as exponential time algorithm for training 1-d GNN, polynomial time algorithm for training GNN on edgeless graphs, and polynomial time algorithm for training a linear GNN.

**Strengths:**

- The main result (Np-hardness of 1-d GNNs) is interesting and uses a non-trivial reduction.

- The supplementary results, provide further insights into the problem.

**Weaknesses:**

- There is no related works section, so it is difficult to position this paper w.r.t previous works. For me, it is difficult for me to determine how novel the result is since I don’t know previous works on the hardness of learning GNNs. It would be helpful to provide a thorough literature survey and more in-depth comparisons to previous works.

- Froese and Hertrich 2023 show that training neural networks is NP-hard even for input dimension 2. Can’t this result be used on GNNs for edgeless graphs using Proposition 2? If so, the contribution of this paper seems incremental as it improves the hardness result from input dimension 2 to input dimension 1.

- The hardness works only for node classification tasks. There should be a discussion about graph-level tasks, which are widely used in practice (perhaps even more than node-level). As a remark, it is OK to focus on node-level tasks, but still, graph-level tasks should be at least discussed on some level.

On the presentation level, the introduction is a bit long and convoluted (almost 3 pages long), which makes it difficult to understand the main message of the paper.

**Questions:**

- How does this work compare to previous works? Specifically, are there any previous works on the hardness of learning GNNs?

- Can the hardness result from Froese and Hertrich 2023 on 2-d networks be transferred to GNNs?

- The depth of the GNN in Theorem 1 isn’t mentioned. Does it work for any depth?

I am willing to reconsider my score based on the author’s response.

---

> ### Author Response · Authors · 2024-11-19
> **Response to Reviewer onPv**
>
> **Answer to Q1 (and W1)**: We would be happy to provide a more comprehensive literature overview for the complexity of training GNNs, however, to the best of our knowledge we are the first to investigate the problem's complexity. This is in stark contrast to the extensive literature on training classical neural networks and we believe that our results represent the first crucial steps towards filling this gap.
>
> **Answer to Q2 (and W2)**: The reduction of Froese and Hertrich establishes the NP-hardness of training neural networks with input dimensionality 2, but their network has a much higher dimensionality (i.e., in the hidden layers) than 2. Hence, translating their result into the GNN setting produces a GNN with much higher dimensionality than 2. To be precise: the complexity of training ReLU-activated classical neural networks of constant dimensionality (both on the input and on each layer) remains an important open question, and one cannot obtain a hardness result similar to ours by reducing from the neural network setting.
>
> **Response to W3**: The revised version now makes it clear that the work considers the classical node classification framework and that examining the graph-level tasks is an important goal for future work. For information about how our results could transfer to graph-level tasks, please see our answer to Q2 of Reviewer tj5Z.
>
> **Answer to Q3**: The depth of the GNN in Theorem 1 is linear in the number of variables of the input SAT instance. While this was previously implicit, we agree that listing it explicitly improves readability and have added it to the 2nd paragraph of the proof.

---

> > ### Comment · Reviewer_onPv · 2024-11-21
> >
> > I thank the authors for the response. I still think that a related works section could greatly benefit the paper. Even if this is the first paper to study NP-hardness for GNNs, there is a vast literature on the computational hardness of neural networks under many different settings. Before the response, it wasn't completely clear to me why previous works didn't cover the results in this paper. Thus, I suggest adding a related works section.
> >
> > However, my concerns are addressed and I don't believe that the absence of a related works section is a reason enough to reject a paper. I updated my score accordingly.

---

### Official Review · Reviewer_Z1QF · 2024-11-01

**Soundness:** 4
**Presentation:** 3
**Contribution:** 3
**Rating:** 6
**Confidence:** 2

**Summary:**

The paper investigated the computational complexity of training GNNs, with a particular focus on the NP-hardness of one-dimensional GNNs, and authors demonstrated that training ReLU-activated GNNs is NP-hard under specific aggregation functions, such as SUM, MEAN, and SPECTRAL Additionally, the paper established upper bounds on algorithmic efficiency and examines how different network architectures affect training efficiency.

**Strengths:**

1. In section 4, the paper clearly constructed a graph that intuitively represents the structure of GNNs, effectively describing the nodes, edges, and the logical relationships associated with Boolean variables, and it also designs several gadgets to ensure the resolution of the SAT problem, providing an excellent insight into the complexity of GNN training.

2.  The paper presented a rigorous theoretical derivation and, in Section 5, established a general algorithmic upper bound for solving ReLU-GNNT, providing an important theoretical framework regarding the complexity and feasibility of training ReLU-GNNT.

3. The paper proved the NP-hardness of training graph neural networks, which provides an important conclusion for exploring the complexity of training GNNs.

**Weaknesses:**

1. In line 308, when constructing the graph in Section 4, the author explained that gray edges and gray dummy vertices are introduced to ensure the nodes have degrees of 2, 4, or 6. What is the rationale behind it? Is it only applicable when the node degree is even, rather than simply being constrained to the specific values of 2, 4, or 6? The author could enhance clarity by providing further explanation on this matter to improve the readability of the paper.

2. The paper researched the NP-Hardness of training ReLU-activated one-dimensional GNNs. The conclusion is primarily correct for the ReLU activation function. If the GNNs' activation function is changed, such as to Sigmoid or Leaky ReLU, the conclusion may be affected. It would be helpful if the authors could provide more theoretical analyses regarding this issue to make the paper more persuasive.

**Questions:**

Refer to wekenesses.

---

> ### Author Response · Authors · 2024-11-19
> **Response to Reviewer Z1QF**
>
> **Response to W1**: We only require the nodes to have degree 2, 4, or 6 because it helps us, later in the proof, to obtain a fully regular graph (i.e. with all nodes having the same degree, in our case precisely 6). Thus, the numbers 2, 4, and 6 at this time of the reduction are not important per se, they simply enable an easier analysis later of how to make our graph 6-regular without changing the way important information propagates over it. We have added some more explanation where these dummy vertices are mentioned in the caption of Fig. 2.
>
> **Response to W2**: We have added a remark about Leaky ReLU and Sigmoid activation functions into the concluding remarks. To provide more context, for Sigmoid the main difficulty is that the research community as a whole lacks the right tools to obtain any complexity-theoretic results at all - even in the conceptually simpler setting of neural network training. For the Leaky ReLU class of functions, in the extremal cases where the Leaky ReLU activation functions coincide with ReLU and linear activation functions respectively, our results carry over immediately, while for others the complexity remains open. In particular, as we do not yet know whether linearly activated GNN training for non-zero error bound is tractable on 1-dimensional GNNs, it is hard to conjecture on the intermediate range of “true” Leaky ReLU activation functions.

---

### Official Review · Reviewer_tj5Z · 2024-11-04

**Soundness:** 4
**Presentation:** 4
**Contribution:** 3
**Rating:** 8
**Confidence:** 4

**Summary:**

The paper studies the computational complexity of graph neural networks, with a focus on one-dimensional GNNs. The primary result is the NP-hardness of training ReLU-activated one-dimensional GNNs. In addition, the paper provides algorithmic upper bounds for the training problem in the ReLU-activated setting, and shows that the one-dimensional, edgeless setting can be solved in polynomial time, as can the linear-activation case.

**Strengths:**

S1. The paper studies an interesting problem that has received little attention in the graph learning community and provides an intriguing result.

S2. The proof technique is innovative, with an elegant reduction based on a non-intuitive connection between a graph learning task and a logic problem.

S3. The paper is well-written. In particular, the results are presented very clearly and proofs are logically structured and readily followed.

**Weaknesses:**

W1. It is challenging to see how the main result extends our understanding of GNNs in an important way. From the perspective of intellectual curiosity, I can appreciate the work as a welcome answer to a question. The proof technique is innovative and elegant. But in most cases, both practical and theoretical (in terms of quantifying the expressive capabilities of a GNN), we are interested in the multi-dimensional setting. Aside from this, the theoretical work focused on pushing the bounds of expressivity has veered away from the simple message-passing approach. The authors don’t provide a clear explanation in the introduction or the conclusion concerning how the presented work is expected to provide further impact.

For many problems, if we want to understand the multi-dimensional setting, then it makes a great deal of sense to first understand the one-dimensional setting. That doesn’t seem to be so clearly the case here, since we already know that the multi-dimensional setting is NP-hard. So how do we gain from deriving special-case results for the one-dimensional setting that can’t be extended to the practically interesting case?

W2. The paper focuses on the (semi-supervised) node classification setting of graph neural networks. While this is clear from Section 2 onwards, it is not mentioned in the abstract or introduction. Graph classification is a very common use case of GNNs, and many of the theoretical results concerning expressivity focus on a GNN’s ability to differentiate between two graphs. The abstract, introduction and limitations section should make the limitations of the derived results much clearer.

**Questions:**

Q1. The paper presents an interesting result and the proof is innovative. On the other hand, as raised in W1, it is challenging to see how this extends our understanding of GNNs in an important way. Could the authors provide an explanation of how they expect the provided results to impact further theoretical work that studies GNNs in the more interesting multi-dimensional setting? Or to the settings that go beyond message-passing (which has known limitations in its expressiveness)? Is there a way to build on the presented proof techniques and use similar concepts for other problems?

Q2. Can the results be extended to the graph classification setting? (Or if it is potentially challenging, can you see paths towards this? Or would it require a totally different approach?)

Q3. Can the authors comment on any connections with their work and the line of research that investigates logical expressiveness (e.g., [R1,R2])? Perhaps there is not an obvious connection, but it would seem that characterization of the types of logical expressions that GNNs are capable of describing has connections to a complexity proof that relies on a linkage to a logic problem. Related to Q1, this branch of work usually assumes a vector at each node, and this is key to expanding the expressive capabilities.

[R1] M. Grohe, "The Descriptive Complexity of Graph Neural Networks," 2023 38th Annual ACM/IEEE Symposium on Logic in Computer Science (LICS), Boston, MA, USA, 2023.

[R2] Barcelo et al., “The Logical Expressiveness of Graph Neural Networks,” in Proc. ICLR, 2020.

---

> ### Author Response · Authors · 2024-11-19
> **Response to Reviewer tj5Z (1/2)**
>
> **Response to W1**: Given the importance and prominence of Graph Neural Networks, we consider it entirely justified to aim for a thorough understanding of the boundaries of (in)tractability of the training problem of GNNs - even if these boundaries do not fall into the settings most used in practical applications. In general, exact polynomial-time algorithms for basic fragments of computational problems can naturally support algorithm design in practical applications by, e.g., providing ideas for heuristics and greedy approaches; however, by establishing intractability even in the one-dimensional case we rule out such an approach when viewed from the natural perspective of dimensionality. Notably, our hardness result demonstrates that the intractability of training GNNs is already due to the difficulty of handling the communication between nodes – and holds even under the simple message-passing approach; hardness does not stem solely from the inherent difficulty of multidimensional classical neural networks training, and hence one cannot hope to make progress by designing heuristics targeting solely the latter aspect. Put more simply: our result can be interpreted as an indication that heuristics for training neural networks cannot be “directly” applied to also work for GNNs. The updated version of the manuscript now also addresses this point in the Concluding Remarks.
>
> It is perhaps worth noting that an assessment of complexity-theoretic lower bounds based on the immediate practical relevance of the results would disqualify most of such bounds that appeared in past editions of ICLR and related conferences (such as ICML, NeurIPS, AAAI and IJCAI). For instance: training neural networks to optimality was long known to be computationally intractable, yet that has not stopped a line of successful research aimed at identifying the precise boundaries of (in)tractability of that fundamental problem (as discussed in the 2nd paragraph of the Introduction).
>
> **Response to W2**: We now make it clear that we study the node classification setting in the abstract and in the third paragraph of the Introduction.
>
> **Answer to Q1**: First of all, prior to our result it was open whether GNN training is polynomial-time tractable for any constant number of dimensions (not only 1). But even beyond this and beyond the context outlined in our response to W1, Theorem 1 shows that attempting to leverage the dimensionality of GNNs to obtain provably efficient algorithms is unlikely to work. This leads to a number of follow-up questions: for instance, if dimensionality is unlikely to lead to tractability on its own, can one use it in combination with the structure of the GNN’s architecture? Or, for two more concrete questions: Do tree-like GNNs (in the sense of having bounded “treewidth”) of low dimensionality admit efficient training? Is the problem tractable on GNN architectures of constant depth and constant dimensionality?
>
> Regarding going beyond the message-passing approach, we believe this can only be done convincingly after one obtains a rigid understanding of the base (but still very challenging) message-passing approach. While it is of course an interesting research direction to study a larger set of node-communication approaches beyond message passing, we see our contribution in just providing the first step by studying the arguably simplest (and yet still prominent) variant. While our hardness results immediately transfer to some variants (e.g., equivariant subgraph aggregation networks [1] for certain subgraph selection policies that merely return the graph itself), others might require different proof techniques and potentially even lead to different complexity landscapes. We updated our conclusion to more explicitly address this field of future work.
>
> **Answer to Q2**: The answer to this question depends on the precise considered graph classification setting. When considering graph classification with a similar propagation procedure and just one read-out at the end (like global-pooling), it seems possible that for some pooling functions our main hardness result can be transferred without altering the construction too much. One crucial obstacle in this is that we require some way to assure that exactly one of the two variable gadgets must have a correctly labeled vertex to properly assign a truth value to the respective variable. If graph classification settings are considered where there is a global exchange of information already in the propagation steps, several of our arguments do not apply anymore. These cases would hence require entirely different or at least heavily adapted arguments.
>
> For the positive result of Theorem 5 on the other hand, it seems quite likely that the ETR instance to which we reduce can be adapted to capture some graph classification settings without too much effort.

---

> ### Author Response · Authors · 2024-11-19
> **Response to Reviewer tj5Z (2/2)**
>
> **Answer to Q3**: In general, there is no direct connection between the complexity of training a GNN and the expressiveness of the logical fragments that can be captured by GNNs. Indeed, the expressiveness of GNNs is inherently tied to the information one can ascertain by evaluating the GNN (with already specified weights and biases), while the training problem aims at determining the weights and biases.
>
> That being said, we are highly appreciative of the work that has been done on linking GNNs to logic. In fact, one of us had actively discussed the training problem with one of the authors of the cited papers after a workshop talk. The outcome - that nothing was known or could be inferred about the GNN training problem - was one of the reasons we set out to fill in this fundamental gap in our understanding.
>
>
> [1] Beatrice Bevilacqua et al. Equivariant subgraph aggregation networks (ICLR 2022)

---

> > ### Comment · Reviewer_tj5Z · 2024-11-22
> > **Acknowledgement of Response**
> >
> > I thank the authors for their thorough response.
> >
> > Concerning the second paragraph in the authors’ response to W1, nowhere in my review did I state that there needs to be "immediate practical relevance" for a theoretical contribution to be worthwhile. The identified weakness W1 did not claim that the result lacked value for this reason. Indeed, the review welcomed the main result and stated that it “studies an interesting problem” and provides “an intriguing result.”
> >
> > The question was really the following: how can other authors build on or learn from the results, either when designing new algorithms or developing new theory?
> >
> > The authors have addressed this satisfactorily in two ways. First, they argue that the presented work demonstrates that the intractability arises from the node communication even in the case of a single dimension, and the work thus establishes that developing heuristics that focus solely on (classical) multidimensional neural network training will not be sufficient in the graph setting. Second, both in the response and the modified conclusion, they identify additional directions to build on the work. The constant depth+dimensionality and bounded treewidth are settings that would be of significant interest.
> >
> > The answers to Q2 and Q3 have helped clarify my understanding. I do think that the extension to graph classification would make for a more complete and powerful work, but it is possibly too much to expect in a single paper. I have increased my score to an accept recommendation.

---

### Author Response · Authors · 2024-11-19

We thank all reviewers for their feedback and constructive comments. Responses to specific questions and concerns are provided to the individual reviews.

---

### Meta-Review · Area_Chair_aH5Z · 2024-12-15

**Metareview:**

The paper investigates the computational complexity of training one-dimensional GNNs, proving NP-hardness for ReLU activation and aggregation functions like sum,mean and spectral. It introduces novel reduction techniques and provides algorithmic upper bounds for simpler cases, such as linear activation and edgeless graphs. The work highlights that intractability arises from node communication rather than dimensionality alone.

strengths: Rigorous techniques address an unexplored area, offering insights into GNN training. The results are well-structured and clearly presented.

weaknesses: fcus is narrow, and the paper lacks discussion on extensions to multi-dimensional and graph-level tasks.

**Additional Comments On Reviewer Discussion:**

Reviewers questioned practical relevance, proof clarity, and generalizability. The authors addressed these effectively by adding illustrative examples (fig 4), clarifying connections to graph-level tasks.

---

### Decision · Program_Chairs · 2025-01-22

Accept (Poster)